# A Machine Learning Model for Predicting Hospitalization in Patients with Respiratory Symptoms during the COVID-19 Pandemic

**DOI:** 10.3390/jcm11154574

**Published:** 2022-08-05

**Authors:** Victor Muniz De Freitas, Daniela Mendes Chiloff, Giulia Gabriella Bosso, Janaina Oliveira Pires Teixeira, Isabele Cristina de Godói Hernandes, Maira do Patrocínio Padilha, Giovanna Corrêa Moura, Luis Gustavo Modelli De Andrade, Frederico Mancuso, Francisco Estivallet Finamor, Aluísio Marçal de Barros Serodio, Jaquelina Sonoe Ota Arakaki, Marair Gracio Ferreira Sartori, Paulo Roberto Abrão Ferreira, Érika Bevilaqua Rangel

**Affiliations:** 1Paulista School of Medicine, Hospital São Paulo, Federal University of São Paulo, São Paulo 04038-901, Brazil; 2Department of Internal Medicine, Botucatu Medical School, University of São Paulo State, Botucatu 18618-687, Brazil; 3Discipline of Emergency Medicine, Department of Medicine, Paulista School of Medicine, Hospital São Paulo, Federal University of São Paulo, São Paulo 04038-901, Brazil; 4Sector of Bioethics, Department of Surgery, Paulista School of Medicine, Hospital São Paulo, Federal University of São Paulo, São Paulo 04038-901, Brazil; 5Pneumology Division, Department of Medicine, Paulista School of Medicine, Hospital São Paulo, Federal University of São Paulo, São Paulo 04038-901, Brazil; 6Department of Obstetrics, Paulista School of Medicine, Hospital São Paulo, Federal University of São Paulo, São Paulo 04038-901, Brazil; 7Infectious Disease Division, Department of Medicine, Paulista School of Medicine, Hospital São Paulo, Federal University of São Paulo, São Paulo 04038-901, Brazil; 8Nephrology Division, Department of Medicine, Paulista School of Medicine, Hospital São Paulo, Federal University of São Paulo, São Paulo 04038-901, Brazil

**Keywords:** machine learning, COVID-19, hospitalization, predictive model

## Abstract

A machine learning approach is a useful tool for risk-stratifying patients with respiratory symptoms during the COVID-19 pandemic, as it is still evolving. We aimed to verify the predictive capacity of a gradient boosting decision trees (XGboost) algorithm to select the most important predictors including clinical and demographic parameters in patients who sought medical support due to respiratory signs and symptoms (RAPID RISK COVID-19). A total of 7336 patients were enrolled in the study, including 6596 patients that did not require hospitalization and 740 that required hospitalization. We identified that patients with respiratory signs and symptoms, in particular, lower oxyhemoglobin saturation by pulse oximetry (SpO_2_) and higher respiratory rate, fever, higher heart rate, and lower levels of blood pressure, associated with age, male sex, and the underlying conditions of diabetes mellitus and hypertension, required hospitalization more often. The predictive model yielded a ROC curve with an area under the curve (AUC) of 0.9181 (95% CI, 0.9001 to 0.9361). In conclusion, our model had a high discriminatory value which enabled the identification of a clinical and demographic profile predictive, preventive, and personalized of COVID-19 severity symptoms.

## 1. Introduction

The coronavirus disease 2019 (COVID-19) posed considerable health pressure on a global scale. In countries with socialized healthcare systems, the sudden influx of patients with respiratory symptoms quickly outgrew the capacity [1,2].

In the early phase of the COVID-19 pandemic, meta-analysis studies documented that dyspnea, anorexia, dizziness, and fatigue were significantly associated with the critical outcome [3]. Likewise, oxyhemoglobin saturation by pulse oximetry (SpO_2_), body temperature, and mean arterial pressure on admission can also predict COVID-19 patients with a high probability of mortality [4]. The COVID-19 timeline was described as the median duration from illness onset to dyspnea of 8.0 days (interquartile range, 5.0–13), whereas from onset of symptoms to first hospital admission was 7.0 days (4.0–8.0) [5,6].

Importantly, the COVID-19 burden is related to underlying comorbidities, in particular hypertension, diabetes mellitus (DM), cardiovascular disease, and chronic kidney disease, which are more likely to be presented with increasing age [7,8]. Other clinical determinants of COVID-19 severity comprise male sex, obesity, smoking, chronic obstructive pulmonary disease (COPD), cerebrovascular disease, malignancy, and chronic liver disease [8].

Therefore, the early identification of factors that predict hospitalization in patients affected by COVID-19 contributes to therapeutic decisions, patient flow management, and allocation of resources. To address these issues, machine learning algorithms allow us to assess comorbidities and clinical features most significantly associated with COVID-19 progression and mortality [9].

In line with these findings, a framework based on male sex, age ≥60 years, known contact with an infected person, and the appearance of five initial clinical symptoms, including cough, fever, headache, sore throat, shortness of breath, enabled the prediction of COVID-19 test results with high accuracy [10]. Importantly, COVID-19 symptoms can be monitored through wireless devices, such as sensors connected to the body and the information can be transmitted through the internet using cell phones. This approach has implications for the prioritization of hospital resources and prevents the spread of SARS-CoV-2 [11]. Moreover, healthcare technology, such as ultrasound and computerized tomography, can also be combined with artificial intelligence to benefit the health system response to the identification of disease clusters, monitoring of cases, mortality risk, COVID-19 diagnosis, disease management, and future pandemic waves of COVID-19 and other respiratory viruses in general [11,12].

Thus, our study aimed to define the main symptoms, signs, and demographic data that were most often associated with the risk of hospitalization in patients with respiratory signs and symptoms that were evaluated in a tertiary hospital during the first wave of COVID-19, from March to August 2020, in São Paulo, Brazil.

## 2. Materials and Methods

### 2.1. Study Population

We reviewed data from all patients referred to the Paulist School of Medicine at Hospital São Paulo, a tertiary university hospital at the Federal University of São Paulo (EPM-UNIFESP), São Paulo, SP, between March 2020 and August 2020 with respiratory syndrome during the first wave of the COVID-19 pandemic. A total of 7336 patients were enrolled in the study individuals, and 6596 patients were not hospitalized, whereas 740 patients required hospitalization. On that occasion, the Unit of Respiratory Infection was temporarily created as a reference for patients with respiratory signs and symptoms during the COVID-19 pandemic.

Exclusion criteria were patients under 18 years old and those who after medical evaluation did not meet the criteria for respiratory disease.

### 2.2. Assessment

We evaluated medical history, and clinical parameters including vital signs, such as blood pressure, respiratory rate, heart rate, temperature, and oxyhemoglobin saturation by pulse oximetry (SpO_2_), demographic data, and pre-existing comorbidities. All data were registered in the electronic health record (EHR). Hospitalization was based on medical decisions, in particular, when SpO_2_ was lower than 94% and respiratory rate was greater than 24 bpm. After hospitalization, more than 90% of the individuals tested positive for SARS-CoV-2 using RT-PCR from nasopharyngeal samples. For those who were not hospitalized, clinical status was monitored by phone calls to retrieve the status (hospitalization or non-hospitalization).

Data were automatically retrieved from the EHR and manually inputted by the researchers. Discrepancies were solved by one of the investigators (V.M.F.) after reviewing the entirety of the patient’s chart. When discrepancies could not be solved or data were not registered, it was marked as “not available”.

From the total of 7336 patients, missing data ranged from 0% to 29.5% of collected information. The least missing data variables were sex and age (no missing data) and the most missing data variables were respiratory rate and seasonal influenza vaccine in the current year (26.6% and 29.5%, respectively) (Appendix A). Regarding comorbidities, data were missing from 11.6% to 11.7% of the total sample, the biggest being active or previous cancer (858 patients) and the least being active smoking (851 patients) (Appendix A). Regarding symptoms, data were missing from 11.8% to 11.9% of the total sample, the biggest being sore throat (871 patients) and the least being sneezing, abdominal pain, and chills (866 patients) (Appendix A).

The Ethics Committee from the Federal University of São Paulo approved the study (CAAE: 41400720.7.0000.5505). All the methods were performed following guidelines and regulations. In addition, this study was carried out under the Declaration of Helsinki. The requirement for informed consent was waived by the Ethics Committee because our study used anonymized data for analysis.

### 2.3. Statistical Analysis

#### 2.3.1. Predictive Model

For the predictive model, the categorical variables were transformed into dummy variables. We removed the variables with more than 30% missing values and imputed the others using nearest neighbors (Appendix A). Once the nearest neighbors are determined, the model is used to predictor nominal variables, and the mean is used for numeric data. Variables with zero or near-zero variance were removed from the model. To fit the boosted tree models, we used one-hot encoding, and to fit the Lasso regression we normalized the predictors.

To adjust to the class imbalance, the synthetic minority over-sampling (SMOTE) method was used to create synthetic classes in the training set (balancing). The SMOTE algorithm generated new examples of the minority class using the nearest neighbors of these cases.

#### 2.3.2. Model Training

We split the data into derivation (training, *n* = 5868) and validation (test, *n* = 1468) data sets. To create the data sets, we used a random split stratified by the target into training (80%) and test (20%). Next, we fitted gradient boosting decision trees (XGBoost), random forest, light GBM, and Lasso regression to all available predictors. The best hyperparameters were selected for each model, using machine learning approaches by 10-fold cross-validation in a train set aimed to maximize the area under the receiver operating characteristic curve (AUC-ROC) and to reduce the bias. Additionally, we fitted a reduced model using the highest scores of the best model (top 11 predictors).

#### 2.3.3. Assessment of Accuracy

The accuracy of the derivation cohort model was tested on the data of the validation cohort. We used the area under the receiver operating characteristic curve (AUC-ROC) and balance accuracy to discriminate the ability of the models in the train and test set. The 95% confidence intervals (CI) of AUC-ROC were estimated by bootstrap resampling (2000 samples) to reduce overfit bias. To evaluate the goodness of fit of models, the predicted versus observed target values were plotted in a confusion matrix of the validation cohort. We select the model with the highest AUC-ROC score and balanced accuracy to fit the reduced model (RAPID RISK COVID-19). The reduced model was fitted using the XGBoost algorithm.

The software R version 4.0.2 (Vienna, Austria) and the packages tidy models.

## 3. Results

In Figure 1, we documented the number of respiratory cases that required hospitalization (*n* = 740) and the cases that presented mild respiratory disease (*n* = 6596) during the first wave of the COVID-19 pandemic at São Paulo Hospital, Federal University of São Paulo, SP, Brazil, from March to August 2020. Therefore, the temporal distribution of evaluated patients shown in Figure 1 reflects the overall tendency registered in Brazil, except for an initial spike at end of March, when restrictive measures were established in São Paulo. During the whole interval evaluated, the number of patients hospitalized remained somewhat constant despite occasional peaks of patients, with an overall hospitalization rate of 10%.

Patients who required hospitalization were more likely to be men with a median age of 58 years (Table 1). The signs which were more often associated with the need for hospitalization comprised lower blood pressure (systolic and diastolic), higher temperature, heart rate, and respiratory rate, whereas oxygen saturation was lower (Table 1). Additionally, we verified that a group of symptoms represented a useful tool for risk-stratifying patients with severe respiratory disease. These symptoms included fatigue, dry cough, breathing difficulty, anorexia, and nausea/vomiting. Conversely, sneezing, running nose, sore throat, headache, and thoracic pain were frequently found in those patients with mild respiratory disease (Table 1). Other symptoms, such as diarrhea, productive cough, anosmia, myalgia, dysgeusia, chills, abdominal pain, and wheezing were equally found in the two groups.

Previous influenza vaccination was observed in those who required hospitalization (Table 1; 47% vs. 39%, *p* = 0.007), which could be explained by the highest age in this group and, therefore, higher access to influenza vaccination before the pandemic. To note, when our study was conducted, there was no report about the therapeutic potential of dexamethasone and remdesivir on this occasion.

When the underlying comorbidities were analyzed, we found that hypertension, cardio- and cerebrovascular diseases, diabetes mellitus (DM), chronic kidney disease, autoimmune disease, neoplasia, obesity, and solid organ transplant were correlated to the need for hospitalization (Table 2). Unexpectedly, asthma, COPD, and smoking were not associated with hospitalization requirements. To note, hospitalization 15 days before seeking medical assistance was reported more often in those who required hospitalization (Table 2; *p* < 0.001).

Next, we split the data into derivation (training, *n* = 5868) and validation (test, *n* = 1468) data sets (Table 3). The models were fitted using all available predictors that were related to symptoms at onset, vital signs at onset, comorbidities, and demography (*n* = 45 predictors). In both data sets, hospitalization was required in 10% of the patients (Table 3). The results showed that boosted trees (XGBoost and light GBM), random forest, and Lasso regression had values of AUC-ROC between 0.899 and 0.930 in the test set (Table 4). To fit the reduced model, we choose the top 11 predictors of the best algorithm. We selected the XGBoost as the best model (higher values of AUC-ROC combined with higher balanced accuracy in the test set).

We then fitted a reduced model using the top 11 predictors retrieved from the XGBoost algorithm, which provided the RAPID RISK COVID-19. The reduced predictive model utilizing SpO_2_, respiratory rate, age, duration of symptoms, sex, temperature at admission, presence of hypertension, heart rate, systolic and diastolic blood pressure at admission, and presence of DM (SHAP, Figure 2A) yielded an AUC-ROC of 0.9181 (95% CI, 0.9001 to 0.9361) (Figure 2B). We showed that the reduced model retrieved similar scores (AUC-ROC, balanced accuracy, precision, and F1-score) compared to the full model (Table 4). To note, SHAP or SHAPley Additive exPlanations is a visualization tool that can be used for making a machine learning model more explainable by visualizing its output. It can be used for explaining the prediction of XGBoost by computing the contribution of each feature to the prediction.

In Figure 3, we summarize our findings.

## 4. Discussion

Through this study, we were able to demonstrate that patients with respiratory signs, in particular, lower SpO_2_ and higher respiratory rate, fever, higher heart rate, and lower levels of blood pressure required hospitalization more often during the first wave of COVID-19 pandemic in a reference university hospital is São Paulo, Brazil. Additionally, age, male sex, longer symptom duration, and comorbidities, such as hypertension and DM, also led to an increase in hospitalization requirements.

Although our study comprised a single center, we observed that our data reflect the pattern of COVID-19 evolution during the first wave of the pandemic in Brazil, when the R_0_ value was estimated at 3.1 [13]. In addition, the state of São Paulo represented the largest state affected by the COVID-19 pandemic, presenting an incidence of 294 hospitalizations per 100,000 inhabitants on that occasion [14]. Most hospitalized patients were reported to be men, elderly, and present underlying comorbidities, including hypertension, chronic lung disease, DM, and cerebrovascular disease [14].

Despite not testing all patients, COVID-19 was the main cause of hospital deaths in 2020 (19.5%), exhibiting an increase of 16.7% in comparison to 2019 in Brazil [15]. Importantly, no influenza pandemic was documented during the current study, so we may assume that we were dealing mainly with cases of COVID-19. As Hospital São Paulo was a reference for respiratory symptoms during the pandemic, our data reflect, therefore, the general population.

Our model (RAPID RISK COVID-19) is a useful tool for risk-stratifying patients with respiratory symptoms, as the COVID-19 pandemic is still evolving. On top of that, a shortage of reagents can difficult the diagnosis of SARS-CoV-2 infection. Therefore, for population health policies, our findings provide evidence to guide those patients who are admitted to a basic health unit or consulted via telemedicine for the need for hospitalization. Notably, peripheral blood oxygenation, heart and respiratory rates, temperature, and blood pressure are inexpensive and easily accessible, as medical professionals can determine health care priorities with real-time assessment. This approach is of paramount importance for identifying high-risk COVID-19 patients, in particular, in low- and middle-income countries.

Lower SpO_2_, higher frequency respiratory rate, fever, fatigue, and cough are important features for the prediction of the COVID-19 severity and were identified as risk factors for hospitalization at the beginning of the pandemic [16,17,18]. These signs may be explained by the direct SARS-CoV-2 infection of the epithelial lung and bronchial cells, as well as, several other organs [19,20], associated with the dysregulation of the renin–angiotensin–aldosterone pathway, which leads to tissue remodeling, inflammation, vasoconstriction, and vascular permeability; endothelial cell damage and thrombo-inflammation, which promotes an imbalance of fibrinolysis and thrombin production, and dysregulated immune response, characterized by T cell lymphopenia due to exhaustion, inhibition of interferon signaling by SARS-CoV-2, and hyperactive innate immunity with higher levels of neutrophils and monocytes, which is ultimately involved in the cytokine storm [20]. SARS-CoV-2 viremia is implicated in worsening COVID-19 [21] and may lead to a systemic response, as documented by higher fever, heart rate, lower blood pressure levels, and fatigue. However, other symptoms, such as a running nose, sore throat, sneezing, headache, and olfactory abnormalities indicate mild cases of COVID-19 [22].

Importantly, machine learning algorithms may also predict the early onset of acute distress respiratory syndrome in critically ill adults with COVID-19 when oxygen saturation, respiratory rate, and blood pressure were evaluated accordingly to age and sex [23]. XGBoost models may also predict COVID-19 progression and mortality when trained with data from the last 24 h. Therefore, respiratory rate, SpO_2_, age greater than 75 years, and laboratory parameters (lactate dehydrogenase, calcium, glucose, and C-reactive protein) were important for risk-stratifying patients with COVID-19 during the first wave, from January to August 2000 [24], which represent the same period when our study was conducted.

To note, using machine learning approaches we fitted machine learning models that achieve higher AUC-ROC values (greater than 0.90) using 45 predictors. The performance of different models was very similar, especially in boosted trees. The models keep a similar performance in the test set that was held separate from the train data indicating a low risk of overfitting. However, in clinical practice, we fitted a reduced model using the top 11 predictors to facilitate the applicability of the algorithm. This model was named RAPID RISK COVID-19 and had performance metrics similar to the complete model in the test set.

In addition, the odds ratio for clinical decompensation within 24 h reaches 5.17 when a SpO_2_ < 93% is associated with leucocytosis and a low glomerular filtration rate is present in patients with heart failure [25]. Lower SpO_2_ values ≤ 90%, measured in an out-of-hospital setting, were associated with a greater than 50% decrease in the probability of being discharged alive, regardless of age [26], supporting, therefore, the decision for prompt hospital admission. The lower levels of PaO_2_/FIO_2_, when associated with lymphopenia, higher levels of C-reactive protein and lactate dehydrogenase, and higher chest tomography scores may also predict prolonged hospitalization [27]. Age, cardiovascular disease, CKD, dyspnoea, tachypnea, confusion, systolic blood pressure, and SpO_2_ ≤ 93% or oxygen requirement discriminate the composite outcome of in-hospital mortality, mechanical ventilation, or admission to the intensive care unit [28]. In a Brazilian and Spanish cohort, seven variables (age, number of comorbidities, heart rate, blood urea nitrogen, C-reactive protein, platelet count, and SpO_2_/FIO_2_) enabled early identification of risk factors predicting in-hospital mortality [29]. SpO_2_ can also be incorporated into the ROX index ([SpO_2_/FIO_2_]/frequency respiratory) to estimate the failure of high-flow cannula failure in COVID-19 patients with acute hypoxemic respiratory failure [30]. Clinicians can take advantage of the results obtained from these models, as an online risk calculator is already available [31].

Age is a risk factor for COVID-19 progression and mortality [7,8,14,22,32,33,34], as aging is frequently associated with the presence of a high number of underlying comorbidities [7] and changes in the immune system including a profile of exaggerated inflammatory response of the innate immune system and immunosenescence of adaptive immune system, which ultimately increases the severity of COVID-19 [35]. Age-related laboratory profile is characterized by higher levels of pro-inflammatory and tissue damage markers, in particular, in elderly men [36]. The lower response to vaccination associated with lower titles of neutralizing antibodies against SARS-CoV-2 [37] poses elderly individuals with a worse outcome.

Male sex is also associated with a higher risk for COVID-19 progression and mortality [14,32,33,38,39]. Sex disparity in COVID-19 severity may be explained by molecular and cellular features. Therefore, females have lower levels of pro-inflammatory and tissue damage markers when compared to males [36,39]. Differences in molecular mechanisms provide a survival advantage in females with COVID-19, as ACE2 and Toll-like receptors (TLR) signaling genes are located in the X chromosome, providing a more versatile and stronger immune response, whereas estrogen contributes to T-cell activation [40]. Therefore, males have a distinct interaction not only between immune cells but also between epithelium and immune cells, which explain the severe outcomes [39].

The presence of underlying comorbidities aggravates the COVID-19 setting. SARS-CoV-2 has a susceptibility to tissue invasion based on the expression of angiotensin-converting enzyme-2 (ACE2) receptors in different organs [19]. Therefore, chronic diseases favor virus entrance and increase the COVID-19 burden [22]. Endothelial damage due to direct virus toxicity and inflammatory response are hallmarks of COVID-19 severity [20,41]. Thus, the presence of cardiometabolic disease, which is another pandemic worldwide, identifies patients with a greater risk of COVID-19 severity. Therefore, hypertension is an independent risk factor for COVID-19 progression and mortality [14,16,18,38]. Blood pressure control is associated with a decrease in the COVID-19 burden and should be encouraged during the pandemic, in patients with advanced atherosclerosis and target organ damage [42].

Likewise, DM is associated with higher rates of mortality due to COVID-19 [14,16,18,32,38]. DM is often associated with hypertension, and cardio- and cerebrovascular diseases, which increase the risk for hospitalization [43]. Recent DM-related pathophysiologic mechanisms in COVID-19 unveiled direct toxicity of islet cells, increase in insulin resistance, and immune system dysfunction associated with microthrombi and endotheliitis, which aggravates glucose control in those patients with pre-existing DM and those with newly diagnosed DM [44]. Adequate blood glucose control ameliorates the COVID-19 burden [45] and should be equally pursued during the pandemic.

Cardiac disease is associated with higher expression of ACE2 and tissue susceptibility to SARS-CoV-2 infection [19], which puts patients with cardiac disease at great risk of hospitalization and mortality [18,25,32,38]. The heart is affected by the inflammatory response, in particular, the cytokine storm and macrophage activation, which promotes endotheliitis, microvascular dysfunction, and thrombosis, in addition to direct damage, an imbalance of ACE2 and angiotensin II, and an increase in myocardial oxygen consumption due to fever, hypoxemia and augmented adrenergic drive [46]. Similarly, cerebrovascular disease can also aggravate COVID-19 [14,34], as SARS-CoV-2 may directly invade neuron cells and also promote endotheliopathy and inflammation [19].

Obesity may also pose COVID-19 patients at high risk for progression [16,32,43], including young patients [47]. Adipose tissue is a reservoir for SARS-CoV-2 replication, as ACE2 is abundantly expressed, which ultimately aggravates the inflammatory milieu [19]. Obese individuals, besides the presence of previous underlying comorbidities, may have a worse pulmonary function performance, presenting lower levels of PaO_2_ and SatO_2_ at admission and the requirement for higher volumes of oxygen [48]. Obese patients express ACE2 more often in the epithelial cells of the lungs [49], which contributes to a worse prognosis.

In line with these findings, chronic kidney disease, which is mainly associated with hypertension, DM, cardio- and cerebrovascular diseases, and obesity, is also correlated to a worse COVID-19 outcome [25,32,43]. SARS-CoV-2 may affect kidney function by several mechanisms, including direct virus toxicity, cytokine storm, rhabdomyolysis, and cardiac decompensation [50].

Solid organ transplant recipients [51] and autoimmune diseases [52] are chronic conditions associated with more severe lymphopenia, a hallmark of COVID-19, and more severe disease, increased viral shedding, lower rates of seroconversion, displaying higher rates of case fatality. Although hematologic solid cancer patients exhibit prolonged viral shedding, delayed or no seroconversion, and sustained immune dysregulation following viral clearance (heterogeneous humoral response and an exhausted T cell phenotype), solid cancer patients present some prolonged viral shedding, early sustained seroconversion, and near-complete resolution of immune dysregulation following viral clearance [53]. Importantly, the need for hospitalization and mortality rates remain higher during all waves of the pandemic, which is a critical condition worldwide and prompts an immediate vaccination regimen for these individuals.

Unexpectedly, smoking and COPD were not correlated to the need for hospitalization, as opposed to the literature [14,18,32]. Both conditions are associated with higher expression of ACE2 in epithelial bronchial and pulmonary cells [19] and systemic inflammatory dysregulation [54]. However, chronic respiratory diseases, yet associated with a higher risk of COVID-19 progression and mortality, were not associated with an augmented risk of hospitalization [55]. The discrepancy in our data may be explained at least in part by the number of patients included in the study and the missing data, which contributed to the under-representation of these patients, or better adherence to precautionary measures.

The impact of asthma on COVID-19 progression is a subject of debate in the literature. Some reports indicate that asthma is an independent risk factor for COVID-19 hospitalization, but not for COVID-19 infection [56]. Our findings showed that pre-existing asthma was not a risk factor for hospitalization, as documented by a meta-analysis [57]. We speculate that better adherence to asthmatic treatments during the pandemic [58], the fact that the gene expression of ACE2, TMPRSS2, and furin is not upregulated in asthmatic patients [59], or that the differences in asthma phenotype that were not deeply analyzed across the studies [60], and the inflammatory milieu in asthmatic patients (type 2 cytokines, including IL-4 and IL-13 and accumulation of eosinophils) or the conventional therapeutics for asthma, including inhaled corticosteroids [61] may have conferred a protective effect for asthmatic patients with COVID-19.

The main limitations of the study were the retrospective analysis, the amount of missing data, and the lack of testing in those patients who were not hospitalized. Additionally, we did not provide external validation in an independent cohort.

In conclusion, while recent advances have been made in the areas of diagnostics, therapeutics, and vaccination for COVID-19, there remains a need for information management tools and a common platform for cross-border collaboration on future pandemic preparedness and response. The lessons learned during the current COVID-19 pandemic have equipped us with a toolbox to tackle future pandemics. Thus, the use of health surveillance technologies [62] associated with the clinical, laboratory, and imaging parameters [12,63,64] will pave the way for the development of predictive, preventive, and personalized solutions.

Therefore, our model had a high discriminatory value that enabled the identification of a clinical and demographic profile predictive of disease severity. Moreover, our approach assisted clinical decision triage and provided additional biological insights into disease progression. However, further research is required to determine whether this tool can also be applied to outpatient or home-based COVID-19 patients, as well as to novel SARS-CoV-2 variants and in the post-vaccination setting.

## Figures and Tables

**Figure 1 jcm-11-04574-f001:**
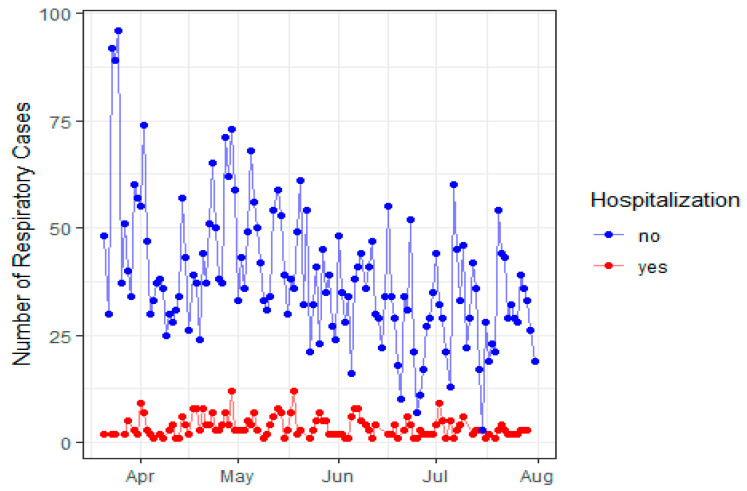
The total number of patients with respiratory signs and symptoms and the need for hospitalization and non-hospitalization, from March to August 2020, at São Paulo Hospital, São Paulo, Brazil.

**Figure 2 jcm-11-04574-f002:**
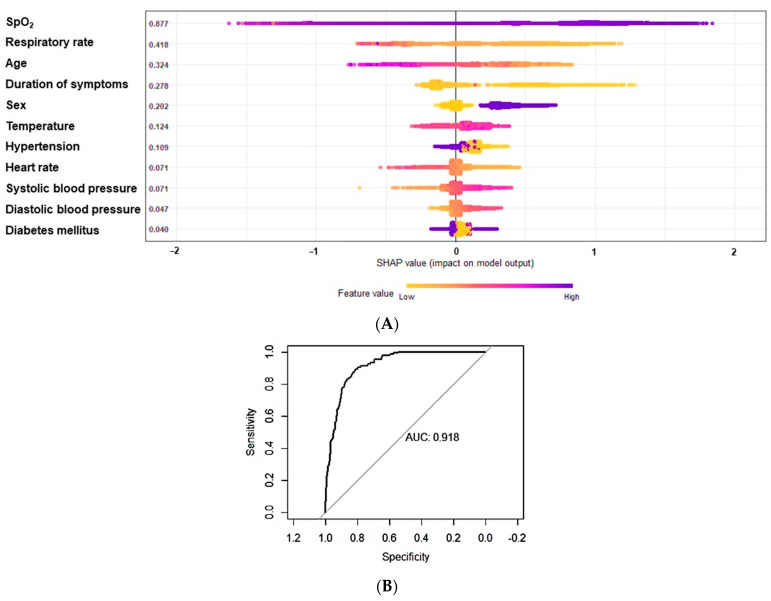
(**A**) Variables in order of importance using the reduced predictive model using SHAP or SHAPley Additive exPlanation plot. (**B**) ROC curve analyses with the area under the curve of 0.9181 (95% CI, 0.9001 to 0.9361).

**Figure 3 jcm-11-04574-f003:**
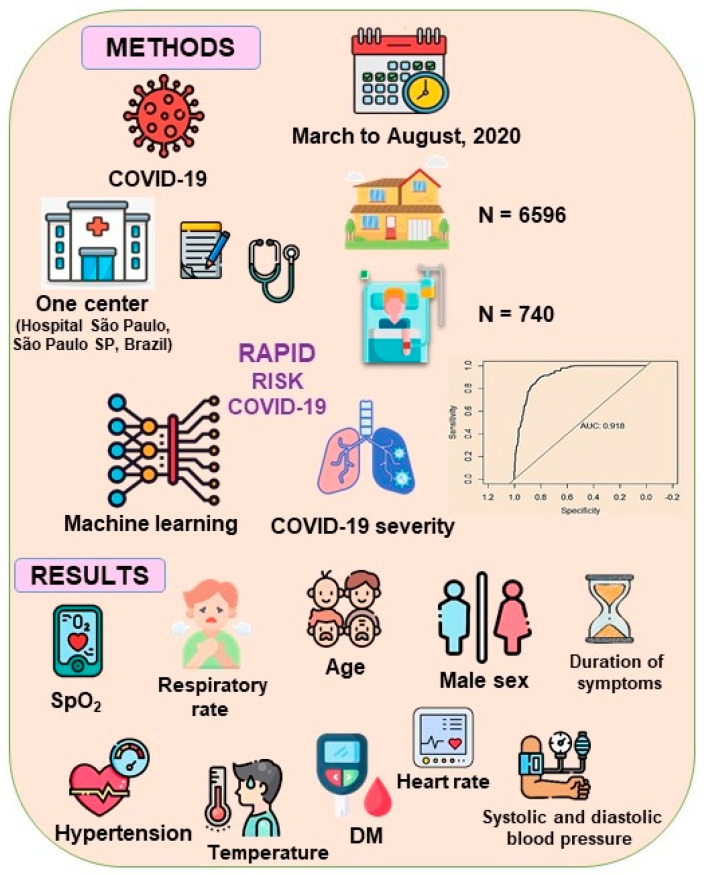
Summary of the study design and findings.

**Table 1 jcm-11-04574-t001:** Comparison of demography, signs, and symptoms between patients who required hospitalization and those who did not require hospitalization from March to August 2020 at São Paulo Hospital, Brazil.

Variables	No Hospitalization (*n* = 6596)		Hospitalization (*n* = 740)		*p*
	Median	Interquartile Range	Median	Interquartile Range	
Females (*n*, %)	3570 (54%)		324 (44%)		<0.001
Age (year-old)	39	(28, 51)	58	(47, 69)	<0.001
Duration of symptoms (days)	4	(2, 8)	7	(4, 10)	<0.001
Systolic blood pressure (mmHg)	133	(121, 146)	129	(114, 145)	<0.001
Diastolic blood pressure (mmHg)	84	(75, 94)	80	(70, 90)	<0.001
Heart rate (bpm)	90	(80, 101)	96	(85, 110)	<0.001
Temperature (°C)	36.50	(36.00, 36.80)	36.50	(36.00, 36.90)	0.007
Respiratory rate (bpm)	18	(16, 20)	24	(20, 28)	<0.001
SpO_2_ (%)	97.00	(96.00, 98.00)	94.00	(90.00, 96.00)	<0.001
Influenza vaccine	1897 (39%)		149 (47%)		0.007
Fever	2500 (42%)		289 (57%)		<0.001
Fatigue	1628 (27%)		184 (37%)		<0.001
Sneezing	496 (8.3%)		20 (4.0%)		<0.001
Dry cough	2702 (45%)		262 (52%)		0.004
Productive cough	720 (12%)		75 (15%)		0.065
Running nose	1304 (22%)		45 (8.9%)		<0.001
Sore throat	1428 (24%)		38 (7.6%)		<0.001
Diarrhea	851 (14%)		88 (17%)		0.051
Breathing difficulty	1955 (33%)		302 (60%)		<0.001
Anorexia	614 (10%)		94 (19%)		<0.001
Headache	2219 (37%)		93 (18%)		<0.001
Myalgia	1845 (31%)		138 (27%)		0.10
Nausea/vomiting	735 (12%)		91 (18%)		<0.001
Wheezing	136 (2.3%)		11 (2.2%)		0.9
Thoracic pain	1059 (18%)		67 (13%)		0.012
Abdominal pain	320 (5.4%)		34 (6.7%)		0.2
Anosmia	1133 (19%)		80 (16%)		0.084
Dysgeusia	1127 (19%)		80 (16%)		0.10
Chills	672 (11%)		46 (9.1%)		0.14

SpO_2_: oxyhemoglobin saturation by pulse oximetry.

**Table 2 jcm-11-04574-t002:** Comparison of comorbidities between patients who required hospitalization and those who did not require hospitalization from March to August 2020 at São Paulo Hospital, Brazil.

Variables	No Hospitalization (*n* = 6596)	Hospitalization (*n* = 740)	*p*
Hypertension	1227 (21%)	232 (46%)	<0.001
Cardiac disease	227 (3.8%)	68 (14%)	<0.001
Diabetes mellitus	469 (7.8%)	136 (27%)	<0.001
Cerebrovascular disease	42 (0.7%)	17 (3.4%)	<0.001
Chronic kidney disease	162 (2.7%)	66 (13%)	<0.001
Immunosuppression	230 (3.8%)	55 (11%)	<0.001
COPD	95 (1.6%)	13 (2.6%)	0.10
Asthma	355 (5.9%)	19 (3.8%)	0.044
Tuberculosis	39 (0.7%)	6 (1.2%)	0.2
Other respiratory diseases	87 (1.5%)	12 (2.4%)	0.10
Neoplasia	89 (1.5%)	30 (6.0%)	<0.001
Solid organ transplant	145 (2.4%)	63 (12%)	<0.001
Obesity	305 (5.1%)	56 (11%)	<0.001
Smoking	621 (10%)	45 (8.9%)	0.3
Pregnancy	79 (1.3%)	3 (0.6%)	0.2
Previous hospitalization	37 (0.6%)	23 (4.5%)	<0.001

COPD: Chronic obstructive pulmonary disease.

**Table 3 jcm-11-04574-t003:** Datasets (train and test sets) used in predicting hospitalization in patients with respiratory symptoms during the COVID-19 pandemic.

Characteristic	Train *n* = 5868	Test *n* = 1468
Female sex	3102 (53%)	792 (54%)
Age	41 (29, 53)	40 (30, 54)
Symptom duration	4 (2, 8)	4 (2, 8)
Systolic blood pressure	133 (121, 147)	132 (120, 145)
Diastolic blood pressure	84 (74, 94)	82 (74, 92)
Heart rate	90 (80, 102)	89 (80, 100)
Temperature	36.50 (36.00, 36.80)	36.50 (36.00, 36.70)
Respiratory frequency	18 (17, 20)	18 (17, 20)
SpO_2_	97.00 (96.00, 98.00)	97.00 (96.00, 98.00)
Influenza vaccine	1632 (40%)	414 (39%)
Fever	2258 (44%)	531 (41%)
Fatigue	1456 (28%)	356 (27%)
Occasional Cough	420 (8.1%)	96 (7.4%)
Dry cough	2370 (46%)	594 (46%)
Phlegm cough	641 (12%)	154 (12%)
Running nose	1059 (20%)	290 (22%)
Sore throat	1173 (23%)	293 (23%)
Diarrhea	733 (14%)	206 (16%)
Dyspnea	1822 (35%)	435 (34%)
Anorexia	580 (11%)	128 (9.9%)
Headache	1851 (36%)	461 (36%)
Myalgia	1597 (31%)	386 (30%)
Nausea and vomiting	652 (13%)	174 (13%)
Chest wheezing	120 (2.3%)	27 (2.1%)
Chest pain	889 (17%)	237 (18%)
Abdominal pain	295 (5.7%)	59 (4.6%)
Anosmia	959 (19%)	254 (20%)
Dysgeusia	960 (19%)	247 (19%)
Chills	570 (11%)	148 (11%)
Hypertension	1193 (23%)	266 (20%)
Heart disease	238 (4.6%)	57 (4.4%)
Diabetes mellitus	466 (9.0%)	139 (11%)
Cerebrovascular disease	47 (0.9%)	12 (0.9%)
Chronic kidney disease	188 (3.6%)	40 (3.1%)
Immunosuppression	231 (4.5%)	54 (4.2%)
COPD	84 (1.6%)	24 (1.8%)
Asthma	307 (5.9%)	67 (5.2%)
Tuberculosis	29 (0.6%)	16 (1.2%)
Respiratory disease	79 (1.5%)	20 (1.5%)
Neoplasia	97 (1.9%)	22 (1.7%)
Transplant	171 (3.3%)	37 (2.8%)
Obesity	284 (5.5%)	77 (5.9%)
Smoking	535 (10%)	131 (10%)
Pregnant	61 (1.2%)	21 (1.6%)
Prior hospitalization	47 (0.9%)	13 (1.0%)
Hospitalization	592 (10%)	148 (10%)

Continuous values are present in medians and percentiles (25 and 75%). SpO_2_: oxyhemoglobin saturation by pulse oximetry; COPD: chronic obstructive pulmonary disease.

**Table 4 jcm-11-04574-t004:** Results of model performance in predicting hospitalization in patients with respiratory symptoms during the COVID-19 pandemic. The predictions were retrieved in the test set (*n* = 1468).

Model	AUC-ROC [95% CI]	Accuracy	Balanced Accuracy	F1-Score	Precision
XGBoost Full ^1^	0.927 [0.901–0.945]	0.905	0.800	0.946	0.962
Random Forest ^1^	0.930 [0.909–0.945]	0.905	0.779	0.947	0.957
Lasso ^1^	0.899 [0.874–0.925]	0.844	0.823	0.907	0.974
Light GBM ^1^	0.925 [0.905–0.944]	0.896	0.822	0.941	0.968
XgBoost Reduced ^2^ (RAPID RISK COVID)	0.917 [0.899–0.935]	0.886	0.793	0.935	0.962

^1^ Predictors of the full model (*n* = 45): sex, age, duration of symptoms, systolic blood pressure, diastolic blood pressure, heart rate, temperature, respiratory frequency, SpO_2_, influenza vaccine, fever, fatigue, occasional cough, dry cough, phlegm cough, running nose, sore throat, diarrhea, dyspnea, anorexia, headache, myalgia, nausea and vomiting, chest wheezing, chest pain, abdominal pain, anosmia, dysgeusia, chills, hypertension, heart disease, diabetes mellitus, cerebrovascular disease, chronic kidney disease, immunosuppression, chronic obstructive pulmonary disease, asthma, tuberculosis, respiratory disease, neoplasia, transplant, obesity, smoking, pregnancy, and prior hospitalization. ^2^ Predictors of reduced model (*n* = 11): SpO_2_, respiratory rate, age, sex, duration of symptoms, presence of hypertension, temperature at admission, presence of DM, heart rate, and systolic and diastolic blood pressure at admission.

## Data Availability

The database is available from the corresponding author upon reasonable request.

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
