# Peer review of "A Machine Learning Model for Predicting Hospitalization in Patients with Respiratory Symptoms during the COVID-19 Pandemic"

_jcm, 2022, doi:10.3390/jcm11154574_

Round 1

Reviewer 1 Report

As per my suggestion, this study has essential content worth publishing. Nevertheless, there is a lot of room for improvement in this paper. Please address the following comments which will further enhance the quality of the manuscript.

1. This paper focuses on COVID-19 and the study is related to that. However, currently, after major vaccinations, COVID-19 is declared ended in many countries. In order to enhance the quality of this manuscript, it is suggested to relate the text about COVID-19 to future pandemics. It is fine to focus on COVID-19 but focusing on future pandemics and relating it to COVID-19 will further increase the quality of the manuscript. There are multiple research articles available online on future pandemics.

2. "A picture is worth a thousand words" - Only two figures are used in this paper. It is suggested to add further figures (if possible) to further improve the quality of the manuscript. Note: Table captions are placed above and figure captions are placed below the figure. Please correct the script accordingly.

3. Introduction part is brief and does not provide sufficient information. It is suggested to add more text in the introduction section related to the focused topic. At least 1 complete page for the introduction part shall be written.

4. Throughout the script, there are grammatical and general language errors. It is advised to run through scripts using sources like Grammarly or any other.

5. Following is an extensive review related to this work. It is suggested to read, understand, and cite the relevant information from the following paper.

[Saeed, U., Shah, S. Y., Ahmad, J., Imran, M. A., Abbasi, Q. H., & Shah, S. A. (2022). Machine learning empowered COVID-19 patient monitoring using non-contact sensing: An extensive review. Journal of pharmaceutical analysis.]

Author Response

We would like to thank the reviewers for their comments. These comments contributed to strengthening our manuscript. We performed substantial changes to the manuscript including (a) the addition of 3 tables (two in the main text and one in the supplementary material); (b) the addition of a new figure to summarize our study; and (c) and the expansion of the Introduction section. All changes are highlighted in yellow throughout the manuscript.   

  1. This paper focuses on COVID-19 and the study is related to that. However, currently, after major vaccinations, COVID-19 is declared ended in many countries. In order to enhance the quality of this manuscript, it is suggested to relate the text about COVID-19 to future pandemics. It is fine to focus on COVID-19 but focusing on future pandemics and relating it to COVID-19 will further increase the quality of the manuscript. There are multiple research articles available online on future pandemics.

Response: Thank you for bringing these topics to discussion. We agree that COVID-19 will not be the last pandemic in our lifetime and future pandemics are inevitable. Therefore, as suggested, we added two paragraphs in the Introduction section and one paragraph in the Discussion section (page 24) to contextualize the importance of our study in the COVID-19 pandemic and future pandemics.        

  1. "A picture is worth a thousand words" - Only two figures are used in this paper. It is suggested to add further figures (if possible) to further improve the quality of the manuscript. Note: Table captions are placed above and figure captions are placed below the figure. Please correct the script accordingly.

Response: Thank you for your suggestion. We added Figure 3 to summarize our findings. We replaced Figure 2A using a SHAP plot. In addition, the captions are placed above tables and below figures, as recommended.   

  1. Introduction part is brief and does not provide sufficient information. It is suggested to add more text in the introduction section related to the focused topic. At least 1 complete page for the introduction part shall be written.

Response: We added more references to the Introduction section and expanded it with a focus on machine learning, the use of healthcare technology to provide information for distinct models, and the importance of our findings in future pandemics.   

  1. Throughout the script, there are grammatical and general language errors. It is advised to run through scripts using sources like Grammarly or any other.

Response: We verified the grammar throughout the manuscript using the Grammarly tool.  

  1. Following is an extensive review related to this work. It is suggested to read, understand, and cite the relevant information from the following paper. [Saeed, U., Shah, S. Y., Ahmad, J., Imran, M. A., Abbasi, Q. H., & Shah, S. A. (2022). Machine learning empowered COVID-19 patient monitoring using non-contact sensing: An extensive review. Journal of pharmaceutical analysis.]. Response: As suggested, we added the reference of Saeed et al.  

Reviewer 2 Report

Thank you for considering me to review this highly important research topic. I would like to commend the authors with their impeccable number of data collected. Some minor revisions are in place to make the paper more applicable and could be cited even in different fields of study. 

From the introduction section, I would suggest the authors to highlight more the related literatures and deeper explanation on the health-related aspect to have a broader range of discussion and attract more readers to cite the paper.

In section 2.2, the missing data was categorized properly, but the total number of dataset were not clearly stated. This should be included as to profoundly explain the datasets available for analysis.

In addition, the processing for detailed data cleaning was not discussed in detail which should be an important step before the data analysis process. 

I suggest the authors to add pseudocodes for the MLA they utilized so other researchers may consider the findings and may extend the study/method.

In the results section, I would like the authors to present the XGBoost classification model since the study highlighted the 'model' for their research contribution which was not present.

The discussion for overfitting/underfitting should be included to assure readers of the accuracy of the results presented.

For comments, why did the authors chose XGBoost rather than other classification model?

Authors were able to discuss and present the contribution quite great. Commend for the work done. I hope that the comments presented may help your paper and future researchers upon reading your work. I enjoyed the application of machine learning in the medical field that you have presented which is both timely and relevant.

Author Response

 We would like to thank the reviewers for their comments. These comments contributed to strengthening our manuscript. We performed substantial changes to the manuscript including (a) the addition of 3 tables (two in the main text and one in the supplementary material); (b) the addition of a new figure to summarize our study; and (c) and the expansion of the Introduction section. All changes are highlighted in yellow throughout the manuscript.   

  • From the introduction section, I would suggest the authors to highlight more the related literatures and deeper explanation on the health-related aspect to have a broader range of discussion and attract more readers to cite the paper.

Response: Thank you for the suggestion. We added more references to the Introduction section and expanded it with a focus on machine learning, the use of healthcare technology to provide information for distinct models, and the importance of our findings in future pandemics.   

  • In section 2.2, the missing data was categorized properly, but the total number of dataset were not clearly stated. This should be included as to profoundly explain the datasets available for analysis.

Response: Thank you for pointing this out.  We added a supplementary table that describes all the predictors and the individual missing values (Table 2S). Additionally, we provided the composition of train and test sets (Table 3).

  • In addition, the processing for detailed data cleaning was not discussed in detail which should be an important step before the data analysis process. 

Response: As suggested, we provided more information on the statistical models, especially in the pre-processing step and the model training (pages 6 &7). 

  • I suggest the authors to add pseudocodes for the MLA they utilized so other researchers may consider the findings and may extend the study/method.

Response: Thank you for this great suggestion. We named the model RAPID RISK COVID-19.

  • In the results section, I would like the authors to present the XGBoost classification model since the study highlighted the 'model' for their research contribution which was not present.

Response: We added more details of the model fitting (pages 12-13 and Tables 3 & 4) and provided another model explanation adding a SHAP plot in Figure 2A (page 14).

  • The discussion for overfitting/underfitting should be included to assure readers of the accuracy of the results presented.

Response: As suggested, we included a topic in the discussion section (2nd paragraph, page 19) and add a limitation of lack of external validation (page 24).

  • For comments, why did the authors chose XGBoost rather than other classification model?

Response: Thank you for bringing this topic up for discussion. We provided the complete approach to model fitting that considered other algorithms. These algorithms were XGBoost, Random Forest, Light GBM, and Lasso regression. We keep the XGBoost because the higher AUC-ROC values combined with higher balanced accuracy. We provided a table (Table 4) in the Results section that showed the model performance metrics in the test set.

  • Authors were able to discuss and present the contribution quite great. Commend for the work done. I hope that the comments presented may help your paper and future researchers upon reading your work. I enjoyed the application of machine learning in the medical field that you have presented which is both timely and relevant.

Response: Thank you very much for your comments. We appreciate them.   

Round 2

Reviewer 1 Report

The author(s) have performed significant revisions. 

Author Response

Thank you for your previous suggestions and for supporting our study.